# Maximizing acquisition functions
# for Bayesian optimization

**James T. Wilson**[*]
Imperial College London

**Frank Hutter**
University of Freiburg

**Marc Peter Deisenroth**
Imperial College London
PROWLER.io

## Abstract

Bayesian optimization is a sample-efficient approach to global optimization that relies on theoretically motivated value heuristics (acquisition functions) to guide its search process. Fully maximizing acquisition functions produces the Bayes' decision rule, but this ideal is difficult to achieve since these functions are frequently non-trivial to optimize. This statement is especially true when evaluating queries in parallel, where acquisition functions are routinely non-convex, high-dimensional, and intractable. We first show that acquisition functions estimated via Monte Carlo integration are consistently amenable to gradient-based optimization. Subsequently, we identify a common family of acquisition functions, including EI and UCB, whose properties not only facilitate but justify use of greedy approaches for their maximization.

## 1 Introduction

Bayesian optimization (BO) is a powerful framework for tackling complicated global optimization problems [32, 40, 44]. Given a black-box function $f : \mathcal{X} \to \mathcal{Y}$, BO seeks to identify a maximizer $\mathbf{x}^* \in \arg\max_{\mathbf{x} \in \mathcal{X}} f(\mathbf{x})$ while simultaneously minimizing incurred costs. Recently, these strategies have demonstrated state-of-the-art results on many important, real-world problems ranging from material sciences [17, 57], to robotics [3, 7], to algorithm tuning and configuration [16, 29, 53, 56].

From a high-level perspective, BO can be understood as the application of Bayesian decision theory to optimization problems [11, 14, 45]. One first specifies a belief over possible explanations for $f$ using a probabilistic surrogate model and then combines this belief with an acquisition function $\mathcal{L}$ to convey the expected utility for evaluating a set of queries $\mathbf{X}$. In theory, $\mathbf{X}$ is chosen according to Bayes' decision rule as $\mathcal{L}$'s maximizer by solving for an *inner optimization problem* [19, 42, 59]. In practice, challenges associated with maximizing $\mathcal{L}$ greatly impede our ability to live up to this standard. Nevertheless, this inner optimization problem is often treated as a black-box unto itself. Failing to address this challenge leads to a systematic departure from BO's premise and, consequently, consistent deterioration in achieved performance.

To help reconcile theory and practice, we present two modern perspectives for addressing BO's inner optimization problem that exploit key aspects of acquisition functions and their estimators. First, we clarify how sample path derivatives can be used to optimize a wide range of acquisition functions estimated via Monte Carlo (MC) integration. Second, we identify a common family of submodular acquisition functions and show that its constituents can generally be expressed in a more computer-friendly form. These acquisition functions' properties enable greedy approaches to efficiently maximize them with guaranteed near-optimal results. Finally, we demonstrate through comprehensive experiments that these theoretical contributions directly translate to reliable and, often, substantial performance gains.

---

[*]Correspondence to j.wilson17@imperial.ac.uk

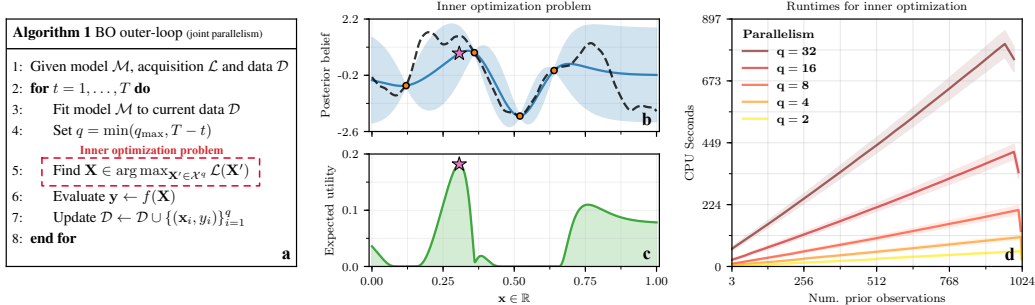

**Figure 1:** (a) Pseudo-code for standard BO's "outer-loop" with parallelism $q$; the inner optimization problem is boxed in red. (b–c) GP-based belief and expected utility (EI), given four initial observations '•'. The aim of the inner optimization problem is to find the optimal query '☆'. (d) Time to compute $2^{14}$ evaluations of MC $q$-EI using a GP surrogate for varied observation counts and degrees of parallelism. Runtimes fall off at the final step because $q$ decreases to accommodate evaluation budget $T = 1,024$.

## 2 Background

Bayesian optimization relies on both a surrogate model $\mathcal{M}$ and an acquisition function $\mathcal{L}$ to define a strategy for efficiently maximizing a black-box function $f$. At each "outer-loop" iteration (Figure 1a), this strategy is used to choose a set of queries $\mathbf{X}$ whose evaluation advances the search process. This section reviews related concepts and closes with discussion of the associated inner optimization problem. For an in-depth review of BO, we defer to the recent survey [52].

Without loss of generality, we assume BO strategies evaluate $q$ designs $\mathbf{X} \in \mathbb{R}^{q \times d}$ in parallel so that setting $q = 1$ recovers purely sequential decision-making. We denote available information regarding $f$ as $\mathcal{D} = \{(\mathbf{x}_i, y_i)\}_{i=1}^{\dots}$ and, for notational convenience, assume noiseless observations $\mathbf{y} = f(\mathbf{X})$. Additionally, we refer to $\mathcal{L}$'s parameters (such as an improvement threshold) as $\psi$ and to $\mathcal{M}$'s parameters as $\zeta$. Henceforth, direct reference to these terms will be omitted where possible.

**Surrogate models** A surrogate model $\mathcal{M}$ provides a probabilistic interpretation of $f$ whereby possible explanations for the function are seen as draws $f^k \sim p(f|\mathcal{D})$. In some cases, this belief is expressed as an explicit ensemble of sample functions [28, 54, 60]. More commonly however, $\mathcal{M}$ dictates the parameters $\theta$ of a (joint) distribution over the function's behavior at a finite set of points $\mathbf{X}$. By first tuning the model's (hyper)parameters $\zeta$ to explain for $\mathcal{D}$, a belief is formed as $p(\mathbf{y}|\mathbf{X}, \mathcal{D}) = p(\mathbf{y}; \theta)$ with $\theta \leftarrow \mathcal{M}(\mathbf{X}; \zeta)$. Throughout, $\theta \leftarrow \mathcal{M}(\mathbf{X}; \zeta)$ is used to denote that belief $p$'s parameters $\theta$ are specified by model $\mathcal{M}$ evaluated at $\mathbf{X}$. A member of this latter category, the Gaussian process prior (GP) is the most widely used surrogate and induces a multivariate normal belief $\theta \triangleq (\mu, \Sigma) \leftarrow \mathcal{M}(\mathbf{X}; \zeta)$ such that $p(\mathbf{y}; \theta) = \mathcal{N}(\mathbf{y}; \mu, \Sigma)$ for any finite set $\mathbf{X}$ (see Figure 1b).

**Acquisition functions** With few exceptions, acquisition functions amount to integrals defined in terms of a belief $p$ over the unknown outcomes $\mathbf{y} = \{y_1, \dots, y_q\}$ revealed when evaluating a black-box function $f$ at corresponding input locations $\mathbf{X} = \{\mathbf{x}_1, \dots, \mathbf{x}_q\}$. This formulation naturally occurs as part of a Bayesian approach whereby the value of querying $\mathbf{X}$ is determined by accounting for the utility provided by possible outcomes $\mathbf{y}^k \sim p(\mathbf{y}|\mathbf{X}, \mathcal{D})$. Denoting the chosen utility function as $\ell$, this paradigm leads to acquisition functions defined as expectations

$$\mathcal{L}(\mathbf{X}; \mathcal{D}, \psi) = \mathbb{E}_{\mathbf{y}}\left[\ell(\mathbf{y}; \psi)\right] = \int \ell(\mathbf{y}; \psi) p(\mathbf{y}|\mathbf{X}, \mathcal{D}) d\mathbf{y}. \tag{1}$$

A seeming exception to this rule, *non-myopic* acquisition functions assign value by further considering how different realizations of $\mathcal{D}_q^k \leftarrow \mathcal{D} \cup \{(\mathbf{x}_i, y_i^k)\}_{i=1}^q$ impact our broader understanding of $f$ and usually correspond to more complex, nested integrals. Figure 1c portrays a prototypical acquisition surface and Table 1 exemplifies popular, myopic and non-myopic instances of (1).

**Inner optimization problem** Maximizing acquisition functions plays a crucial role in BO as the process through which abstract machinery (e.g. model $\mathcal{M}$ and acquisition function $\mathcal{L}$) yields concrete actions (e.g. decisions regarding sets of queries $\mathbf{X}$). Despite its importance however, this inner optimization problem is often neglected. This lack of emphasis is largely attributable to a greater

| Abbr. | Acquisition Function $\mathcal{L}$ | Reparameterization | MM |
|---|---|---|---|
| EI | $\mathbb{E}_{\mathbf{y}}[\max(\text{ReLU}(\mathbf{y} - \alpha))]$ | $\mathbb{E}_{\mathbf{z}}[\max(\text{ReLU}(\boldsymbol{\mu} + \mathbf{L}\mathbf{z} - \alpha))]$ | Y |
| PI | $\mathbb{E}_{\mathbf{y}}[\max(\mathbb{1}^-(\mathbf{y} - \alpha))]$ | $\mathbb{E}_{\mathbf{z}}[\max(\sigma(\frac{\boldsymbol{\mu}+\mathbf{L}\mathbf{z}-\alpha}{\tau}))]$ | Y |
| SR | $\mathbb{E}_{\mathbf{y}}[\max(\mathbf{y})]$ | $\mathbb{E}_{\mathbf{z}}[\max(\boldsymbol{\mu} + \mathbf{L}\mathbf{z})]$ | Y |
| UCB | $\mathbb{E}_{\mathbf{y}}[\max(\boldsymbol{\mu} + \sqrt{\beta\pi/2}\,\lvert\boldsymbol{\gamma}\rvert)]$ | $\mathbb{E}_{\mathbf{z}}[\max(\boldsymbol{\mu} + \sqrt{\beta\pi/2}\,\lvert\mathbf{L}\mathbf{z}\rvert)]$ | Y |
| ES | $-\mathbb{E}_{\mathbf{y}_a}[\text{H}(\mathbb{E}_{\mathbf{y}_b\mid\mathbf{y}_a}[\mathbb{1}^+(\mathbf{y}_b - \max(\mathbf{y}_b))])]$ | $-\mathbb{E}_{\mathbf{z}_a}[\text{H}(\mathbb{E}_{\mathbf{z}_b}[\text{softmax}(\frac{\boldsymbol{\mu}_{b\mid a}+\mathbf{L}_{b\mid a}\mathbf{z}_b}{\tau})])]$ | N |
| KG | $\mathbb{E}_{\mathbf{y}_a}[\max(\boldsymbol{\mu}_b + \boldsymbol{\Sigma}_{b,a}\boldsymbol{\Sigma}_{a,a}^{-1}(\mathbf{y}_a - \boldsymbol{\mu}_a))]$ | $\mathbb{E}_{\mathbf{z}_a}[\max(\boldsymbol{\mu}_b + \boldsymbol{\Sigma}_{b,a}\boldsymbol{\Sigma}_{a,a}^{-1}\mathbf{L}_a\mathbf{z}_a)]$ | N |

**Table 1:** Examples of reparameterizable acquisition functions; the final column indicates whether they belong to the MM family (Section 3.2). Glossary: $\mathbb{1}^{+/-}$ denotes the right-/left-continuous Heaviside step function; ReLU and $\sigma$ rectified linear and sigmoid nonlinearities, respectively; H the Shannon entropy; $\alpha$ an improvement threshold; $\tau$ a temperature parameter; $\mathbf{L}\mathbf{L}^\top \triangleq \boldsymbol{\Sigma}$ the Cholesky factor; and, residuals $\boldsymbol{\gamma} \sim \mathcal{N}(\mathbf{0}, \boldsymbol{\Sigma})$. Lastly, non-myopic acquisition function (ES and KG) are assumed to be defined using a discretization. Terms associated with the query set and discretization are respectively denoted via subscripts $a$ and $b$.

focus on creating new and improved machinery as well as on applying BO to new types of problems. Moreover, elementary examples of BO facilitate $\mathcal{L}$'s maximization. For example, optimizing a single query $\mathbf{x} \in \mathbb{R}^d$ is usually straightforward when $\mathbf{x}$ is low-dimensional and $\mathcal{L}$ is myopic.

Outside these textbook examples, however, BO's inner optimization problem becomes qualitatively more difficult to solve. In virtually all cases, acquisition functions are non-convex (frequently due to the non-convexity of plausible explanations for $f$). Accordingly, increases in input dimensionality $d$ can be prohibitive to efficient query optimization. In the generalized setting with parallelism $q \geq 1$, this issue is exacerbated by the additional scaling in $q$. While this combination of non-convexity and (acquisition) dimensionality is problematic, the routine intractability of both non-myopic and parallel acquisition poses a commensurate challenge.

As is generally true of integrals, the majority of acquisition functions are intractable. Even Gaussian integrals, which are often preferred because they lead to analytic solutions for certain instances of (1), are only tractable in a handful of special cases [13, 18, 20]. To circumvent the lack of closed-form solutions, researchers have proposed a wealth of diverse methods. Approximation strategies [13, 15, 60], which replace a quantity of interest with a more readily computable one, work well in practice but may not to converge to the true value. In contrast, bespoke solutions [10, 20, 22] provide (near-)analytic expressions but typically do not scale well with dimensionality.[2] Lastly, MC methods [27, 47, 53] are highly versatile and generally unbiased, but are often perceived as non-differentiable and, therefore, inefficient for purposes of maximizing $\mathcal{L}$.

Regardless of the method however, the (often drastic) increase in cost when evaluating $\mathcal{L}$'s proxy acts as a barrier to efficient query optimization, and these costs increase over time as shown in Figure 1d. In an effort to address these problems, we now go inside the outer-loop and focus on efficient methods for maximizing acquisition functions.

## 3 Maximizing acquisition functions

This section presents the technical contributions of this paper, which can be broken down into two complementary topics: 1) gradient-based optimization of acquisition functions that are estimated via Monte Carlo integration, and 2) greedy maximization of "myopic maximal" acquisition functions. Below, we separately discuss each contribution along with its related literature.

### 3.1 Differentiating Monte Carlo acquisitions

Gradients are one of the most valuable sources of information for optimizing functions. In this section, we detail both the reasons and conditions whereby MC acquisition functions are differentiable and further show that most well-known examples readily satisfy these criteria (see Table 1).

We assume that $\mathcal{L}$ is an expectation over a multivariate normal belief $p(\mathbf{y}|\mathbf{X}, \mathcal{D}) = \mathcal{N}(\mathbf{y}; \boldsymbol{\mu}, \boldsymbol{\Sigma})$ specified by a GP surrogate such that $(\boldsymbol{\mu}, \boldsymbol{\Sigma}) \leftarrow \mathcal{M}(\mathbf{X})$. More generally, we assume that samples can be generated as $\mathbf{y}^k \sim p(\mathbf{y}|\mathbf{X}, \mathcal{D})$ to form an unbiased MC estimator of an acquisition function $\mathcal{L}(\mathbf{X}) \approx \mathcal{L}_m(\mathbf{X}) \triangleq \frac{1}{m} \sum_{k=1}^{m} \ell(\mathbf{y}^k)$. Given such an estimator, we are interested in verifying whether

$$\nabla\mathcal{L}(\mathbf{X}) \approx \nabla\mathcal{L}_m(\mathbf{X}) \triangleq \frac{1}{m} \sum_{k=1}^{m} \nabla\ell(\mathbf{y}^k), \tag{2}$$

where $\nabla\ell$ denotes the gradient of utility function $\ell$ taken with respect to $\mathbf{X}$. The validity of MC gradient estimator (2) is obscured by the fact that $\mathbf{y}^k$ depends on $\mathbf{X}$ through generative distribution $p$ and that $\nabla\mathcal{L}_m$ is the expectation of $\ell$'s derivative rather than the derivative of its expectation.

Originally referred to as *infinitesimal perturbation analysis* [8, 24], the *reparameterization trick* [37, 50] is the process of differentiating through an MC estimate to its generative distribution $p$'s parameters and consists of two components: i) reparameterizing samples from $p$ as draws from a simpler base distribution $\hat{p}$, and ii) interchanging differentiation and integration by taking the expectation over sample path derivatives.

**Reparameterization** Reparameterization is a way of interpreting samples that makes their differentiability w.r.t. a generative distribution's parameters transparent. Often, samples $\mathbf{y}^k \sim p(\mathbf{y}; \boldsymbol{\theta})$ can be re-expressed as a deterministic mapping $\phi : \mathcal{Z} \times \boldsymbol{\Theta} \rightarrow \mathcal{Y}$ of simpler random variates $\mathbf{z}^k \sim \hat{p}(\mathbf{z})$ [37, 50]. This change of variables helps clarify that, if $\ell$ is a differentiable function of $\mathbf{y} = \phi(\mathbf{z}; \boldsymbol{\theta})$, then $\frac{d\ell}{d\boldsymbol{\theta}} = \frac{d\ell}{d\phi}\frac{d\phi}{d\boldsymbol{\theta}}$ by the chain rule of (functional) derivatives.

If generative distribution $p$ is multivariate normal with parameters $\boldsymbol{\theta} = (\boldsymbol{\mu}, \boldsymbol{\Sigma})$, the corresponding mapping is then $\phi(\mathbf{z}; \boldsymbol{\theta}) \triangleq \boldsymbol{\mu} + \mathbf{L}\mathbf{z}$, where $\mathbf{z} \sim \mathcal{N}(\mathbf{0}, \mathbf{I})$ and $\mathbf{L}$ is $\boldsymbol{\Sigma}$'s Cholesky factor such that $\mathbf{L}\mathbf{L}^\top = \boldsymbol{\Sigma}$. Rewriting (1) as a Gaussian integral and reparameterizing, we have

$$\mathcal{L}(\mathbf{X}) = \int_{\boldsymbol{a}}^{\boldsymbol{b}} \ell(\mathbf{y})\mathcal{N}(\mathbf{y}; \boldsymbol{\mu}, \boldsymbol{\Sigma})d\mathbf{y} = \int_{\boldsymbol{a}'}^{\boldsymbol{b}'} \ell(\boldsymbol{\mu} + \mathbf{L}\mathbf{z})\mathcal{N}(\mathbf{z}; \mathbf{0}, \mathbf{I})d\mathbf{z}, \tag{3}$$

where each of the $q$ terms $c_i'$ in both $\boldsymbol{a}'$ and $\boldsymbol{b}'$ is transformed as $c_i' = (c_i - \mu_i - \sum_{j<i} L_{ij}z_j)/L_{ii}$. The third column of Table 1 grounds (3) with several prominent examples. For a given draw $\mathbf{y}^k \sim \mathcal{N}(\boldsymbol{\mu}, \boldsymbol{\Sigma})$, the sample path derivative of $\ell$ w.r.t. $\mathbf{X}$ is then

$$\nabla\ell(\mathbf{y}^k) = \frac{d\ell(\mathbf{y}^k)}{d\mathbf{y}^k}\frac{d\mathbf{y}^k}{d\mathcal{M}(\mathbf{X})}\frac{d\mathcal{M}(\mathbf{X})}{d\mathbf{X}}, \tag{4}$$

where, by minor abuse of notation, we have substituted in $\mathbf{y}^k = \phi(\mathbf{z}^k; \mathcal{M}(\mathbf{X}))$. Reinterpreting $\mathbf{y}$ as a function of $\mathbf{z}$ therefore sheds light on individual MC sample's differentiability.

**Interchangeability** Since $\mathcal{L}_m$ is an unbiased MC estimator consisting of differentiable terms, it is natural to wonder whether the average sample gradient $\nabla\mathcal{L}_m$ (2) follows suit, i.e. whether

$$\nabla\mathcal{L}(\mathbf{X}) = \nabla\mathbb{E}_{\mathbf{y}}[\ell(\mathbf{y})] \stackrel{?}{=} \mathbb{E}_{\mathbf{y}}[\nabla\ell(\mathbf{y})] \approx \nabla\mathcal{L}_m(\mathbf{X}), \tag{5}$$

where $\stackrel{?}{=}$ denotes a potential equivalence when interchanging differentiation and expectation. Necessary and sufficient conditions for this interchange are that, as defined under $p$, integrand $\ell$ must be continuous and its first derivative $\ell'$ must a.s. exist and be integrable [8, 24]. Wang et al. [59] demonstrated that these conditions are met for a GP with a twice differentiable kernel, provided that the elements in query set $\mathbf{X}$ are unique. The authors then use these results to prove that (2) is an unbiased gradient estimator for the parallel Expected Improvement ($q$-EI) acquisition function [10, 22, 53]. In later works, these findings were extended to include parallel versions of the Knowledge Gradient (KG) acquisition function [61, 62]. Figure 2d (bottom right) visualizes gradient-based optimization of MC $q$-EI for parallelism $q = 2$.

**Extensions** Rather than focusing on individual examples, our goal is to show differentiability for a broad class of MC acquisition functions. In addition to its conceptual simplicity, one of MC integration's primary strengths is its generality. This versatility is evident in Table 1, which catalogs (differentiable) reparameterizations for six of the most popular acquisition functions. While some

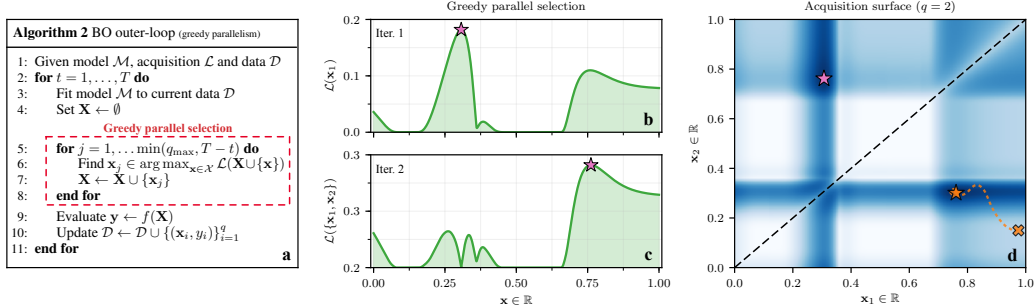

**Figure 2:** (a) Pseudo-code for BO outer-loop with greedy parallelism, the inner optimization problem is boxed in red. (b–c) Successive iterations of greedy maximization, starting from the posterior shown in Figure 1b. (d) On the left, greedily selected query '☆'; on the right and from '×' to '☆', trajectory when jointly optimizing parallel queries $\mathbf{x}_1$ and $\mathbf{x}_2$ via stochastic gradient ascent. Darker colors correspond with larger acquisitions.

of these forms were previously known (EI and KG) or follow freely from the above (SR), others require additional steps. We summarize these steps below and provide full details in Appendix A.

In many cases of interest, utility is measured in terms of discrete events. For example, Probability of Improvement [40, 58] is the expectation of a binary event $e_{\text{PI}}$: "will a new set of results improve upon a level $\alpha$?" Similarly, Entropy Search [27] contains expectations of categorical events $e_{\text{ES}}$: "which of a set of random variables will be the largest?" Unfortunately, mappings from continuous variables $\mathbf{y}$ to discrete events $e$ are typically discontinuous and, therefore, violate the conditions for (5). To overcome this issue, we utilize *concrete* (continuous to discrete) approximations in place of the original, discontinuous mappings [31, 41].

Still within the context of the reparameterization trick, [31, 41] studied the closely related problem of optimizing an expectation w.r.t. a discrete generative distribution's parameters. To do so, the authors propose relaxing the mapping from, e.g., uniform to categorical random variables with a continuous approximation so that the (now differentiable) transformed variables closely resemble their discrete counterparts in distribution. Here, we first map from uniform to Gaussian (rather than Gumbel) random variables, but the process is otherwise identical. Concretely, we can approximate PI's binary event as

$$\tilde{\boldsymbol{e}}_{\text{PI}}(\mathbf{X}; \alpha, \tau) = \max\left(\sigma\left(\mathbf{y} - \alpha/\tau\right)\right) \approx \max\left(\mathbb{1}^-(\mathbf{y} - \alpha)\right), \tag{6}$$

where $\mathbb{1}^-$ denotes the left-continuous Heaviside step function, $\sigma$ the sigmoid nonlinearity, and $\tau \in [0, \infty]$ acts as a temperature parameter such that the approximation becomes exact as $\tau \to 0$. Appendix A.1 further discusses concrete approximations for both PI and ES.

Lastly, the Upper Confidence Bound (UCB) acquisition function [55] is typically not portrayed as an expectation, seemingly barring the use of MC methods. At the same time, the standard definition $\text{UCB}(\mathbf{x}; \beta) \triangleq \mu + \beta^{1/2}\sigma$ bares a striking resemblance to the reparameterization for normal random variables $\phi(z; \mu, \sigma) = \mu + \sigma z$. By exploiting this insight, it is possible to rewrite this closed-form expression as $\text{UCB}(\mathbf{x}; \beta) = \int_\mu^\infty y \mathcal{N}(y; \mu, 2\pi\beta\sigma^2) dy$. Formulating UCB as an expectation allows us to naturally parallelize this acquisition function as

$$\text{UCB}(\mathbf{X}; \beta) = \mathbb{E}_{\mathbf{y}}\left[\max(\boldsymbol{\mu} + \sqrt{\beta\pi/2}|\boldsymbol{\gamma}|)\right], \tag{7}$$

where $|\boldsymbol{\gamma}| = |\mathbf{y} - \boldsymbol{\mu}|$ denotes the absolute value of $\mathbf{y}$'s residuals. In contrast with existing parallelizations of UCB [12, 15], Equation (7) directly generalizes its marginal form and can be efficiently estimated via MC integration (see Appendix A.2 for the full derivation).

These extensions further demonstrate how many of the apparent barriers to gradient-based optimization of MC acquisition functions can be overcome by borrowing ideas from new (and old) techniques.

## 3.2 Maximizing myopic maximal acquisitions

This section focuses exclusively on the family of myopic maximal (MM) acquisition functions: myopic acquisition functions defined as the expected max of a pointwise utility function $\hat{\ell}$, i.e.

$\mathcal{L}(\mathbf{X}) = \mathbb{E}_{\mathbf{y}}[\ell(\mathbf{y})] = \mathbb{E}_{\mathbf{y}}[\max \hat{\ell}(\mathbf{y})]$. Of the acquisition functions included in Table 1, this family includes EI, PI, SR, and UCB. We show that these functions have special properties that make them particularly amenable to greedy maximization.

Greedy maximization is a popular approach for selecting near-optimal sets of queries $\mathbf{X}$ to be evaluated in parallel [1, 9, 12, 15, 35, 51]. This iterative strategy is so named because it always "greedily" chooses the query $\mathbf{x}$ that produces the largest immediate reward. At each step $j = 1, \ldots, q$, a greedy maximizer treats the $j-1$ preceding choices $\mathbf{X}_{<j}$ as constants and grows the set by selecting an additional element $\mathbf{x}_j \in \arg \max_{\mathbf{x} \in \mathcal{X}} \mathcal{L}(\mathbf{X}_{<j} \cup \{\mathbf{x}\}; \mathcal{D})$ from the set of possible queries $\mathcal{X}$. Algorithm 2 in Figure 2 outlines this process's role in BO's outer-loop.

**Submodularity**  Greedy maximization is often linked to the concept of *submodularity* (SM). Roughly speaking, a set function $\mathcal{L}$ is SM if its increase in value when adding any new point $\mathbf{x}_j$ to an existing collection $\mathbf{X}_{<j}$ is non-increasing in cardinality $k$ (for a technical overview, see [2]). Greedily maximizing SM functions is guaranteed to produce near-optimal results [39, 43, 46]. Specifically, if $\mathcal{L}$ is a normalized SM function with maximum $\mathcal{L}^*$, then a greedy maximizer will incur no more than $\frac{1}{e}\mathcal{L}^*$ regret when attempting to solve for $\mathbf{X}^* \in \arg\max_{\mathbf{X} \in \mathcal{X}^q} \mathcal{L}(\mathbf{X})$.

In the context of BO, SM has previously been appealed to when establishing outer-loop regret bounds [12, 15, 55]. Such applications of SM utilize this property by relating an idealized BO strategy to greedy maximization of a SM objective (e.g., the mutual information between black-box function $f$ and observations $\mathcal{D}$). In contrast, we show that the family of MM acquisition functions are inherently SM, thereby guaranteeing that greedy maximization thereof produces near-optimal choices $\mathbf{X}$ at each step of BO's outer-loop.[3] We begin by removing some unnecessary complexity:

1. Let $f^k \sim p(f|\mathcal{D})$ denote the $k$-th possible explanation of black-box $f$ given observations $\mathcal{D}$. By marginalizing out nuisance variables $f(\mathcal{X} \setminus \mathbf{X})$, $\mathcal{L}$ can be expressed as an expectation over functions $f^k$ themselves rather than over potential outcomes $\mathbf{y}^k \sim p(\mathbf{y}|\mathbf{X}, \mathcal{D})$.

2. Belief $p(f|\mathcal{D})$ and sample paths $f^k$ depend solely on $\mathcal{D}$. Hence, expected utility $\mathcal{L}(\mathbf{X}; \mathcal{D}) = \mathbb{E}_f [\ell(f(\mathbf{X}))]$ is a weighted sum over a fixed set of functions whose weights are constant. Since non-negative linear combinations of SM functions are SM [39], $\mathcal{L}(\cdot)$ is SM so long as the same can be said of all functions $\ell(f^k(\cdot)) = \max \hat{\ell}(f^k(\cdot))$.

3. As pointwise functions, $f^k$ and $\hat{\ell}$ specify the set of values mapped to by $\mathcal{X}$. They therefore influences whether we can normalize the utility function such that $\ell(\emptyset) = 0$, but do not impact SM. Appendix A.3 discusses the technical condition of normalization in greater detail. In general however, we require that $v_{\min} = \min_{\mathbf{x} \in \mathcal{X}} \hat{\ell}(f^k(\mathbf{x}))$ is guaranteed to be bounded from below for all functions under the support of $p(f|\mathcal{D})$.

Having now eliminated confounding factors, the remaining question is whether $\max(\cdot)$ is SM. Let $\mathcal{V}$ be the set of possible utility values and define $\max(\emptyset) = v_{\min}$. Then, given sets $\mathcal{A} \subseteq \mathcal{B} \subseteq \mathcal{V}$ and $\forall v \in \mathcal{V}$, it holds that

$$\max(\mathcal{A} \cup \{v\}) - \max(\mathcal{A}) \geq \max(\mathcal{B} \cup \{v\}) - \max(\mathcal{B}). \tag{8}$$

*Proof:* We prove the equivalent definition $\max(\mathcal{A}) + \max(\mathcal{B}) \geq \max(\mathcal{A} \cup \mathcal{B}) + \max(\mathcal{A} \cap \mathcal{B})$. Without loss of generality, assume $\max(\mathcal{A} \cup \mathcal{B}) = \max(\mathcal{A})$. Then, $\max(\mathcal{B}) \geq \max(\mathcal{A} \cap \mathcal{B})$ since, for any $\mathcal{C} \subseteq \mathcal{B}$, $\max(\mathcal{B}) \geq \max(\mathcal{C})$.

This result establishes the MM family as a class of SM set functions, providing strong theoretical justification for greedy approaches to solving BO's inner-optimization problem.

**Incremental form**  So far, we have discussed greedy maximizers that select a $j$-th new point $\mathbf{x}_j$ by optimizing the joint acquisition $\mathcal{L}(\mathbf{X}_{1:j}; \mathcal{D}) = \mathbb{E}_{\mathbf{y}_{1:j}|\mathcal{D}} [\ell(\mathbf{y}_{1:j})]$ originally defined in (1). A closely related strategy [12, 15, 23, 53] is to formulate the greedy maximizer's objective as (the expectation of) a marginal acquisition function $\bar{\mathcal{L}}$. We refer to this category of acquisition functions, which explicitly represent the value of $\mathbf{X}_{1:j}$ as that of $\mathbf{X}_{<j}$ incremented by a marginal quantity, as *incremental*. The most common example of an incremental acquisition function is the iterated

expectation $\mathbb{E}_{\mathbf{y}_{<j}|\mathcal{D}}\left[\bar{\mathcal{L}}(\mathbf{x}_j;\mathcal{D}_j)\right]$, where $\mathcal{D}_j = \mathcal{D} \cup \{(\mathbf{x}_i, y_i)\}_{i<j}$ denotes a fantasy state. Because these integrals are generally intractable, MC integration (Section 3.1) is typically used to estimate their values by averaging over fantasies formed by sampling from $p(\mathbf{y}_{<j}|\mathbf{X}_{<j}, \mathcal{D})$.

In practice, approaches based on incremental acquisition functions (such as the mentioned MC estimator) have several distinct advantages over joint ones. Marginal (myopic) acquisition functions usually admit differentiable, closed-form solutions. The latter property makes them cheap to evaluate, while the former reduces the sample variance of MC estimators. Moreover, these approaches can better utilize caching since many computationally expensive terms (such as a Cholesky used to generate fantasies) only change between rounds of greedy maximization.

A joint acquisition function $\mathcal{L}$ can always be expressed as an incremental one by defining $\bar{\mathcal{L}}$ as the expectation of the corresponding utility function $\ell$'s discrete derivative

$$\Delta(\mathbf{x}_j; \mathbf{X}_{<j}, \mathcal{D}) = \mathbb{E}_{\mathbf{y}_{1:j}|\mathcal{D}}\left[\delta(y_j; \mathbf{y}_{<j})\right] = \mathcal{L}(\mathbf{X}_{1:j}; \mathcal{D}) - \mathcal{L}(\mathbf{X}_{<j}; \mathcal{D}), \tag{9}$$

with $\delta(y_j; \mathbf{y}_{<j}) = \ell(\mathbf{y}_{1:j}) - \ell(\mathbf{y}_{<j})$ and $\mathcal{L}(\emptyset; \mathcal{D}) = 0$ so that $\mathcal{L}(\mathbf{X}_{1:q}; , \mathcal{D}) = \sum_{j=1}^q \Delta(\mathbf{x}_j; \mathbf{X}_{<j}, \mathcal{D})$. To show why this representation is especially useful for MM acquisition functions, we reuse the notation of (8) to introduce the following straightforward identity

$$\max(\mathcal{B}) - \max(\mathcal{A}) = \mathrm{ReLU}\left(\max(\mathcal{B} \setminus \mathcal{A}) - \max(\mathcal{A})\right). \tag{10}$$

*Proof:* Since $v_{\min}$ is defined as the smallest possible element of either set, the ReLU's argument is negative if and only if $\mathcal{B}$'s maximum is a member of $\mathcal{A}$ (in which case both sides equate to zero). In all other cases, the ReLU can be eliminated and $\max(\mathcal{B}) = \max(\mathcal{B} \setminus \mathcal{A})$ by definition.

Reformulating the MM marginal gain function as $\delta(y_j; \mathbf{y}_{<j}) = \mathrm{ReLU}(\ell(y_j) - \ell(\mathbf{y}_{<j}))$ now gives the desired result: that the MM family's discrete derivative is the "improvement" function. Accordingly, the conditional expectation of (9) given fantasy state $\mathcal{D}_j$ is the expected improvement of $\ell$, i.e.

$$\mathbb{E}_{y_j|\mathcal{D}_j}\left[\delta(y_j; \mathbf{y}_{<j})\right] = \mathrm{EI}_\ell\left(\mathbf{x}_j; \mathcal{D}_j\right) = \int_{\Gamma_j} \left[\ell(y_j) - \ell(\mathbf{y}_{<j})\right] p(y_j|\mathbf{x}_j, \mathcal{D}_j) dy_j, \tag{11}$$

where $\Gamma_j \triangleq \{y_j : \ell(y_j) > \ell(\mathbf{y}_{<j})\}$. Since marginal gain function $\delta$ primarily acts to lower bound a univariate integral over $y_j$, (11) often admits closed-form solutions. This statement is true of all MM acquisition functions considered here, making their incremental forms particularly efficient.

Putting everything together, an MM acquisition function's joint and incremental forms equate as $\mathcal{L}(\mathbf{X}_{1:q}; \mathcal{D}) = \sum_{j=1}^q \mathbb{E}_{\mathbf{y}_{<j}|\mathcal{D}}\left[\mathrm{EI}_\ell\left(\mathbf{x}_j; \mathcal{D}_j\right)\right]$. For the special case of Expected Improvement per se (denoted here as $\mathcal{L}_{\mathrm{EI}}$ to avoid confusion), this expression further simplifies to reveal an exact equivalence whereby $\mathcal{L}_{\mathrm{EI}}(\mathbf{X}_{1:q}; \mathcal{D}) = \sum_{j=1}^q \mathbb{E}_{\mathbf{y}_{<j}|\mathcal{D}}\left[\mathcal{L}_{\mathrm{EI}}(\mathbf{x}_j; \mathcal{D}_j)\right]$. Appending B.3 compares performance when using joint and incremental forms, demonstrating how the latter becomes increasingly beneficial as the dimensionality of the (joint) acquisition function $q \times d$ grows.

## 4 Experiments

We assessed the efficacy of gradient-based and submodular strategies for maximizing acquisition function in two primary settings: "synthetic", where task $f$ was drawn from a known GP prior, and "black-box", where $f$'s nature is unknown to the optimizer. In both cases, we used a GP surrogate with a constant mean and an anisotropic Matérn-⁵⁄₂ kernel. For black-box tasks, ambiguity regarding the correct function prior was handled via online MAP estimation of the GP's (hyper)parameters. Appendix B.1 further details the setup used for synthetic tasks.

We present results averaged over 32 independent trials. Each trial began with three randomly chosen inputs, and competing methods were run from identical starting conditions. While the general notation of the paper has assumed noise-free observations, all experiments were run with Gaussian measurement noise leading to observed values $\hat{y} \sim \mathcal{N}(f(\mathbf{x}), 1\mathrm{e}{-}3)$.

**Acquisition functions** We focused on parallel MC acquisition functions $\mathcal{L}_m$, particularly EI and UCB. Results using EI are shown here and those using UCB are provided in extended results (Appendix B.3). To avoid confounding variables when assessing BO performance for different acquisition maximizers, results using the incremental form of $q$-EI discussed in Section 3.2 are also reserved for extended results.

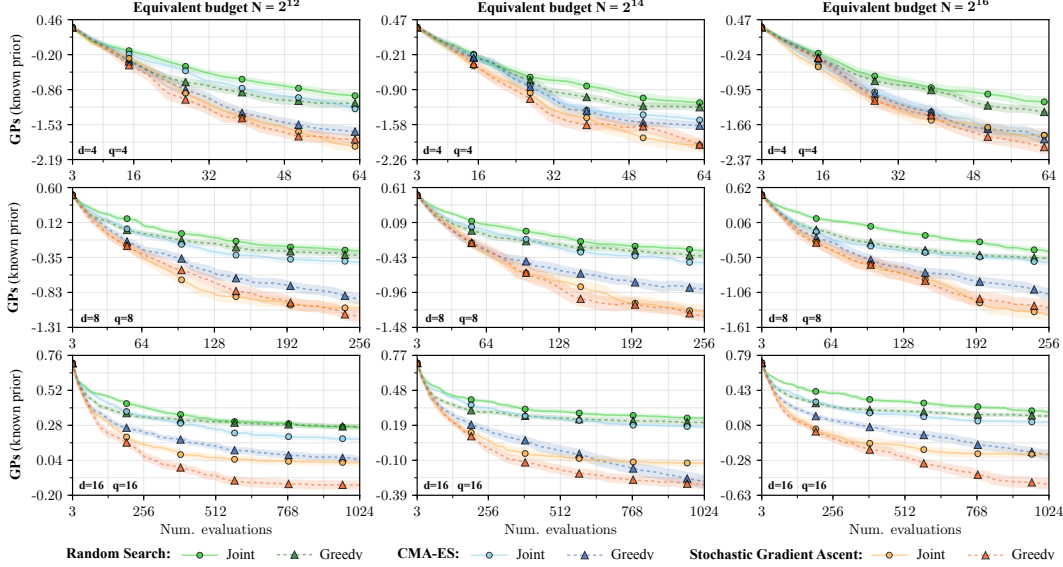

**Figure 3:** Average performance of different acquisition maximizers on synthetic tasks from a known prior, given varied runtimes when maximizing Monte Carlo $q$-EI. Reported values indicate the log of the immediate regret $\log_{10}|f_{\max} - f(\mathbf{x}^*)|$, where $\mathbf{x}^*$ denotes the observed maximizer $\mathbf{x}^* \in \arg\max_{\mathbf{x}\in\mathcal{D}} \hat{y}$.

In additional experiments, we observed that optimization of PI and SR behaved like that of EI and UCB, respectively. However, overall performance using these acquisition functions was slightly worse, so further results are not reported here. Across experiments, the $q$-UCB acquisition function introduced in Section 3.1 outperformed $q$-EI on all tasks but the Levy function.

Generally speaking, MC estimators $\mathcal{L}_m$ come in both deterministic and stochastic varieties. Here, determinism refers to whether or not each of $m$ samples $\mathbf{y}^k$ were generated using the same random variates $\mathbf{z}^k$ within a given outer-loop iteration (see Section 3.1). Together with a decision regarding "batch-size" $m$, this choice reflects a well-known tradeoff of approximation-, estimation-, and optimization-based sources of error when maximizing the true function $\mathcal{L}$ [6]. We explored this tradeoff for each maximizer and summarize our findings below.

**Maximizers** We considered a range of (acquisition) maximizers, ultimately settling on stochastic gradient ascent (ADAM, [36]), Covariance Matrix Adaptation Evolution Strategy (CMA-ES, [26]) and Random Search (RS, [4]). Additional information regarding these choices is provided in Appendix B.1. For fair comparison, maximizers were constrained by CPU runtime. At each outer-loop iteration, an "inner budget" was defined as the average time taken to simultaneously evaluate $N$ acquisition values given equivalent conditions. When using greedy parallelism, this budget was split evenly among each of $q$ iterations. To characterize performance as a function of allocated runtime, experiments were run using inner budgets $N \in \{2^{12}, 2^{14}, 2^{16}\}$.

For ADAM, we used stochastic minibatches consisting of $m = 128$ samples and an initial learning rate $\eta = 1/40$. To combat non-convexity, gradient ascent was run from a total of 32 (64) starting positions when greedily (jointly) maximizing $\mathcal{L}$. Appendix B.2 details the multi-start initialization strategy. As with the gradient-based approaches, CMA-ES performed better when run using stochastic minibatches ($m = 128$). Furthermore, reusing the aforementioned initialization strategy to generate CMA-ES's initial population of 64 samples led to additional performance gains.

**Empirical results** Figures 3 and 4 present key results regarding BO performance under varying conditions. Both sets of experiments explored an array of input dimensionalities $d$ and degrees of parallelism $q$ (shown in the lower left corner of each panel). Maximizers are grouped by color, with darker colors denoting use of greedy parallelism; inner budgets are shown in ascending order from left to right.

Results on synthetic tasks (Figure 3), provide a clearer picture of the maximizers' impacts on the full BO loop by eliminating the model mismatch. Across all dimensions $d$ (rows) and inner budgets

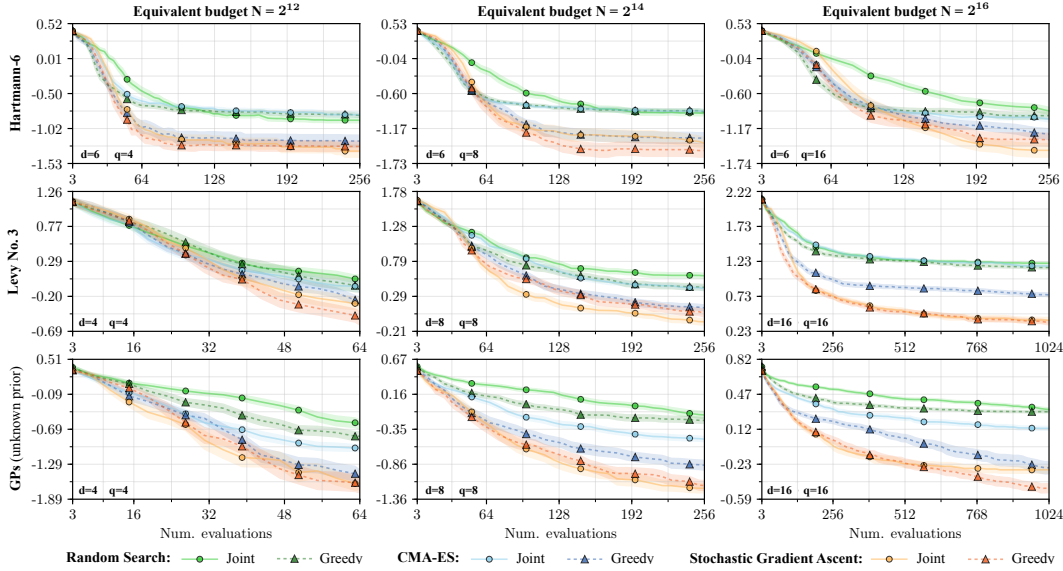

**Figure 4:** Average performance of different acquisition maximizers on black-box tasks from an unknown prior, given varied runtimes when maximizing Monte Carlo $q$-EI. Reported values indicate the log of the immediate regret $\log_{10}|f_{\max} - f(\mathbf{x}^*)|$, where $\mathbf{x}^*$ denotes the observed maximizer $\mathbf{x}^* \in \arg\max_{\mathbf{x} \in \mathcal{D}} \hat{y}$.

$N$ (columns), gradient-based maximizers (orange) were consistently superior to both gradient-free (blue) and naïve (green) alternatives. Similarly, submodular maximizers generally surpassed their joint counterparts. However, in lower-dimensional cases where gradients alone suffice to optimize $\mathcal{L}_m$, the benefits for coupling gradient-based strategies with near-optima seeking submodular maximization naturally decline. Lastly, the benefits of exploiting gradients and submodularity both scaled with increasing acquisition dimensionality $q \times d$.

Trends are largely identical for black-box tasks (Figure 4), and this commonality is most evident for tasks sampled from an unknown GP prior (final row). These runs were identical to ones on synthetic tasks (specifically, the diagonal of Figure 3) but where knowledge of $f$'s prior was withheld. Outcomes here clarify the impact of model mismatch, showing how maximizers maintain their influence. Finally, performance on Hartmann-6 (top row) serves as a clear indicator of the importance for thoroughly solving the inner optimization problem. In these experiments, performance improved despite mounting parallelism due to a corresponding increase in the inner budget.

Overall, these results clearly demonstrate that both gradient-based and submodular approaches to (parallel) query optimization lead to reliable and, often, substantial improvement in outer-loop performance. Furthermore, these gains become more pronounced as the acquisition dimensionality increases. Viewed in isolation, maximizers utilizing gradients consistently outperform gradient-free alternatives. Similarly, greedy strategies improve upon their joint counterparts in most cases.

## 5   Conclusion

BO relies upon an array of powerful tools, such as surrogate models and acquisition functions, and all of these tools are sharpened by strong usage practices. We extend these practices by demonstrating that Monte Carlo acquisition functions provide unbiased gradient estimates that can be exploited when optimizing them. Furthermore, we show that many of the same acquisition functions form a family of submodular set functions that can be efficiently optimized using greedy maximization. These insights serve as cornerstones for easy-to-use, general-purpose techniques for practical BO. Comprehensive empirical evidence concludes that said techniques lead to substantial performance gains in real-world scenarios where queries must be chosen in finite time. By tackling the inner optimization problem, these advances directly benefit the theory and practice of Bayesian optimization.

## Acknowledgments

The authors thank David Ginsbourger, Dario Azzimonti and Henry Wynn for initial discussions regarding the submodularity of various integrals. The support of the EPSRC Centre for Doctoral Training in High Performance Embedded and Distributed Systems (reference EP/L016796/1) is gratefully acknowledged. This work has partly been supported by the European Research Council (ERC) under the European Union's Horizon 2020 research and innovation programme under grant no. 716721.

## Footnotes

[2] By *near-analytic*, we refer to cases where an expression contains terms that cannot be computed exactly but for which high-quality solvers exist (e.g. low-dimensional multivariate normal CDF estimators [20, 21]).

[3]An additional technical requirement for SM is that the ground set $\mathcal{X}$ be finite. Under similar conditions, SM-based guarantees have been extended to infinite ground sets [55], but we have not yet taken these steps.

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
