[Supplementary Material]

# A  Methods Appendix

## A.1  Concrete approximations

**Figure 5:** Left: Concrete approximation to PI for temperatures $\tau \in [1e{-}3, 1e{-}1]$. Middle: Concrete approximation to ES for temperatures $\tau \in [1e{-}3, 1e{-}1]$. Right: Stochastic gradient ascent trajectories when maximizing concrete approximation to parallel versions of PI (top left) and ES (bottom right), both with a temperature $\tau = 0.01$.

As per Section 3.1, utility is sometimes measured in terms of discrete events $e \in \mathcal{E}$. Unfortunately, mappings from continuous values $\mathbf{y} \in \mathcal{Y}$ to the space of discrete events $\mathcal{E}$ are typically discontinuous and, therefore, violate a necessary condition for interchangeability of differentiation and expectation. In the generalized context of differentiating a Monte Carlo integral w.r.t. the parameters of a *discrete* generative distribution, [31, 41] proposed to resolve this issue by introducing a continuous approximation to the aforementioned discontinuous mapping.

As a guiding example, assume that $\boldsymbol{\theta}$ is a self-normalized vector of $q$ parameters such that $\forall \theta \in \boldsymbol{\theta}, \theta \geq 0$ and that $\mathbf{z} = [z_1, \ldots, z_q]^\top$ is a corresponding vector of uniform random variables. Subsequently, let $\phi(\mathbf{z}; \boldsymbol{\theta}) = \log(-\boldsymbol{\theta}/\log \mathbf{z})$ be defined as random variables $\mathbf{y}$'s reparameterization. Denoting by $y^* = \max(\mathbf{y})$, the vector-valued function $\boldsymbol{e} : \mathcal{Y}^q \mapsto \{0,1\}^q$ defined as

$$\boldsymbol{e}(\mathbf{z}; \boldsymbol{\theta}) = [y_1 \geq y^*, \ldots, y_q \geq y^*]^\top = \mathbb{1}^+ (\mathbf{y} - y^*) \tag{12}$$

then reparameterizes a (one-hot encoded) categorical random variable $\boldsymbol{e}$ having distribution $p(\boldsymbol{e}; \boldsymbol{\theta}) = \prod_{i=1}^q \theta_i^{e_i}$. Importantly, we can rewrite (12) as the zero-temperature limit of as continuous mapping $\tilde{\boldsymbol{e}} : \mathcal{Y}^q \mapsto [0,1]^q$ defined as

$$\tilde{\boldsymbol{e}}(\mathbf{y}; \tau) = \text{softmax}\left(\frac{\mathbf{y} - y^*}{\tau}\right) = \text{softmax}\left(\frac{\mathbf{y}}{\tau}\right), \tag{13}$$

where $\tau \in [0, \infty]$ is a temperature parameter. For non-zero temperatures $\tau > 0$, we obtain a relaxed version of the original (one-hot encoded) categorical event. Unlike the original however, the relaxed event satisfies the conditions for interchanging differentiation and expectation.

Returning to the case of an acquisition function (1) defined over a multivariate normal belief $p(\mathbf{y}|\mathbf{X})$ with parameters $\boldsymbol{\theta} = (\boldsymbol{\mu}, \boldsymbol{\Sigma})$, random variables $\mathbf{y}$ are instead reparameterized by $\phi(\mathbf{z}; \theta) = \boldsymbol{\mu} + \mathbf{L}\,\text{g}(\mathbf{z})$, where g denotes, e.g., the Box-Muller transform of uniform rvs $\mathbf{z} = [z_1, \ldots, z_{2q}]^\top$. This particular example demonstrates how Entropy Search's innermost integrand can be relaxed using a *concrete* approximation. Identical logic can be applied to approximate Probability of Improvement's integrand.

For Monte Carlo versions of both acquisition functions, Figure 5 shows the resulting approximation across a range of temperatures along with gradient-based optimization in the parallel setting $q = 2$. Whereas for high temperatures $\tau$ the approximations wash out, both converge to the corresponding true function as $\tau \to 0$.

## A.2 Parallel Upper Confidence Bound ($q$-UCB)

For convenience, we begin by reproducing (3) as indefinite integrals,

$$\mathcal{L}(\mathbf{X}) = \int_{-\infty}^{\infty} \ell(\mathbf{y})\mathcal{N}(\mathbf{y}; \boldsymbol{\mu}, \boldsymbol{\Sigma})d\mathbf{y} = \int_{-\infty}^{\infty} \ell(\boldsymbol{\mu} + \mathbf{Lz})\mathcal{N}(\mathbf{z}; \mathbf{0}, \mathbf{I})d\mathbf{z}.$$

Working backward through this equation, we derive an exact expression for parallel UCB. To this end, we introduce the definition

$$\sqrt{\frac{\pi}{2}} \int_{-\infty}^{\infty} |\sigma z|\mathcal{N}(z; 0, 1)dz = \sqrt{2\pi} \int_{0}^{\infty} y\mathcal{N}(y; 0, \sigma^2)dy = \sigma, \tag{14}$$

where $|\cdot|$ denotes the (pointwise) absolute value operator.[4] Using this fact and given $z \sim \mathcal{N}(0, 1)$, let $\hat{\sigma}^2 \triangleq (\beta\pi/2)\sigma^2$ such that $\mathbb{E}|\hat{\sigma}z| = \beta^{1/2}\sigma$. Under this notation, marginal UCB can be expressed as

$$\begin{aligned} \text{UCB}(\mathbf{x}; \beta) &= \mu + \beta^{1/2}\sigma \\ &= \int_{-\infty}^{\infty} \left(\mu + |\hat{\sigma}z|\right)\mathcal{N}(z; 0, 1)dz \\ &= \int_{-\infty}^{\infty} \left(\mu + |\gamma|\right)\mathcal{N}(\gamma; 0, \hat{\sigma}^2)d\gamma \end{aligned} \tag{15}$$

where $(\mu, \sigma^2)$ parameterize a Gaussian belief over $y = f(\mathbf{x})$ and $\gamma = y - \mu$ denotes $y$'s residual. This integral form of UCB is advantageous precisely because it naturally lends itself to the generalized expression

$$\begin{aligned} \text{UCB}(\mathbf{X}; \beta) &= \int_{-\infty}^{\infty} \max(\boldsymbol{\mu} + |\boldsymbol{\gamma}|)\mathcal{N}(\boldsymbol{\gamma}; \mathbf{0}, \hat{\boldsymbol{\Sigma}})d\boldsymbol{\gamma} \\ &= \int_{-\infty}^{\infty} \max(\boldsymbol{\mu} + |\hat{\mathbf{L}}\mathbf{z}|)\mathcal{N}(\mathbf{z}; \mathbf{0}, \mathbf{I})d\mathbf{z} \\ &\approx \frac{1}{m}\sum_{k=1}^{m} \max(\boldsymbol{\mu} + |\hat{\mathbf{L}}\mathbf{z}^k|) \text{ for } \mathbf{z}^k \sim \mathcal{N}(\mathbf{0}, \mathbf{I}), \end{aligned} \tag{16}$$

where $\hat{\mathbf{L}}\hat{\mathbf{L}}^\top = \hat{\boldsymbol{\Sigma}} \triangleq (\beta\pi/2)\boldsymbol{\Sigma}$. This representation has the requisite property that, for any size $q' \leq q$ subset of $\mathbf{X}$, the value obtained when marginalizing out the remaining $q - q'$ terms is its $q'$-UCB value.

Previous methods for parallelizing UCB have approached the problem by imitating a purely sequential strategy [12, 15]. Because a fully Bayesian approach to sequential selection generally involves an exponential number of posteriors, these works incorporate various well-chosen heuristics for the purpose of efficiently approximate parallel UCB.[5] By directly addressing the associated $q$-dimensional integral however, Eq. (16) avoids the need for such approximations and, instead, unbiasedly estimates the true value.

Finally, the special case of marginal UCB (15) can be further simplified as

$$\text{UCB}(\mathbf{x}; \beta) = \mu + 2\int_{0}^{\infty} \hat{\sigma}z\mathcal{N}(z; 0, 1)dz = \int_{\mu}^{\infty} y\mathcal{N}(y; \mu, 2\pi\beta\sigma^2)dy, \tag{17}$$

revealing an intuitive form — namely, the expectation of a Gaussian random variable (with rescaled covariance) above its mean.

### A.3 Normalizing utility functions

An additional requirement when proving the near-optimality of greedy maximization for a SM function $\mathcal{L}$ is that $\mathcal{L}$ be a normalized set function such that $\mathcal{L}(\emptyset) = 0$. As in Section 3.2, let $v_{\min}$ be defined as the smallest possible utility value given a utility function $\ell$ defined over a ground set $f^k$ indexed by $\mathcal{X}$. Because the max is additive such that $\max(\mathbf{y} - v_{\min}) = \max(\mathbf{y}) - v_{\min}$, normalization is only necessary when establishing regret bounds and simply requires lower bounding $v_{\min} > -\infty$. This task is is facilitated by the fact that $v_{\min}$ pertains to the outputs of $\ell$ rather than to (a belief over) black-box $f$. Addressing the matter by case, we have:

a. Expected Improvement: For a given threshold $\alpha$, let improvement be defined (pointwise) as $\mathrm{ReLU}(\mathbf{y} - \alpha) = \max(0, \mathbf{y} - \alpha)$. EI's integrand is then the largest improvement $\ell_{\mathrm{EI}}(\mathbf{y}; \alpha) = \max(\mathrm{ReLU}(\mathbf{y} - \alpha))$.[6] Applying the rectifier prior to the max defines $\ell_{\mathrm{EI}}$ as a normalized, submodular function.

b. Probability of Improvement: PI's integrand is defined as $\ell_{\mathrm{PI}}(\mathbf{y}, \alpha) = \max(\mathbb{1}^-(\mathbf{y} - \alpha))$, where $\mathbb{1}^-$ denotes the left-continuous Heaviside step function. Seeing as the Heaviside maps $\mathcal{Y} \mapsto \{0, 1\}$, $\ell_{\mathrm{PI}}$ is already normalized.

c. Simple Regret: The submodularity of Simple Regret was previously discussed in [1], under the assumption $v_{\min} = 0$. More generally, normalizing $\ell_{\mathrm{SR}}$ requires bounding $f$'s infimum under $p$. Technical challenges for doing so make submodular maximization of SR the hardest to justify.

d. Upper Confidence Bound: As per (7), define UCB's integrand as the maximum over $\mathbf{y}$'s expectation incremented by non-negative terms. By definition then, $\ell_{\mathrm{UCB}}$ is lower bounded by the predictive mean and can therefore be normalized as $\bar{\ell}_{\mathrm{UCB}} = \max(\boldsymbol{\mu} + |\boldsymbol{\gamma}| - v_{\min})$, provided that $v_{\min} = \min_{\mathbf{x} \in \mathcal{X}} \mu(\mathbf{x})$ is finite. For a zero-mean GP with a twice differentiable kernel, this condition is guaranteed for bounded functions $f$.

### A.4 Expected Improvement's incremental form

For the special case of $\mathcal{L}_{\mathrm{EI}}$, the expected improvement of improvement integrand $\ell_{\mathrm{EI}}$ simplifies as:

$$
\begin{aligned}
\mathrm{EI}_{\ell_{\mathrm{EI}}}(\mathbf{x}_j, \mathcal{D}_j) &= \mathbb{E}_{y_j} \left[ \mathrm{ReLU} \left( \mathrm{ReLU}(y_j - \alpha) - \max \mathrm{ReLU}(\mathbf{y}_{<j} - \alpha) \right) \right] \\
&= \mathbb{E}_{y_j} \left[ \mathrm{ReLU} \left( \max(\alpha, y_j) - \max(\alpha, \max \mathbf{y}_{<j}) \right) \right] \\
&= \mathbb{E}_{y_j} \left[ \mathrm{ReLU} \left( y_j - \max(\alpha, \max \mathbf{y}_{<j}) \right) \right] \\
&= \mathcal{L}_{\mathrm{EI}}(\mathbf{x}_j; \mathcal{D}_j),
\end{aligned}
\tag{18}
$$

where $\alpha = \max(\{y : \forall (\mathbf{x}, y) \in \mathcal{D}\})$ denotes the initial improvement threshold.

## B  Experiments Appendix

### B.1 Experiment Details

**Synthetic tasks**   To eliminate model error, experiments were first run on synthetic tasks drawn from a known prior. For a GP with a continuous, stationary kernel, approximate draws $f$ can be constructed via a weighted sum of basis functions sampled from the corresponding Fourier dual [5, 28, 48, 49]. For a Matérn-$\nu/2$ kernel with anisotropic lengthscales $\boldsymbol{\Lambda}^{-1}$, the associated spectral density is the multivariate $t$-distribution $t_\nu(\mathbf{0}, \boldsymbol{\Lambda}^{-1})$ with $\nu$ degrees of freedom [38]. For our experiments, we set $\boldsymbol{\Lambda} = (d/16)\, \mathbf{I}$ and approximated $f$ using $2^{14}$ basis functions, resulting in tasks that were sufficiently challenging and closely resembled exact draws from the prior.

**Maximizers**   In additional to findings reported in the text, we compared several gradient-based approaches (incl. L-BFGS-B and Polyak averaging [59]) and found that ADAM consistently delivered superior performance. CMA-ES was included after repeatedly outperforming the rival black-box method DIRECT [33]. RS was chosen as a naïve acquisition maximizer after Successive Halving (SH, [30, 34]) failed to yield significant improvement. For consistency when identifying the best proposed query set(s), both RS and SH used deterministic estimators $\mathcal{L}_m$. Whereas RS was run

with a constant batch-size $m = 1024$, SH started small and iteratively increased $m$ to refine estimated acquisition values for promising candidates using a cumulative moving average and cached posteriors.

## B.2 Multi-start initialization

As noted in [59], gradient-based query optimization strategies are often sensitive to the choice of starting positions. This sensitivity naturally occurs for two primary reasons. First, acquisition functions $\mathcal{L}$ are consistently non-convex. As a result, it is easy for members of query set $\mathbf{X} \subseteq \mathcal{X}$ to get stuck in local regions of the space. Second, acquisition surfaces are frequently patterned by (large) plateaus offering little expected utility. Such plateaus typically emerge when corresponding regions of $\mathcal{X}$ are thought to be inferior and are therefore excluded from the search process.

To combat this issue, we appeal to the submodularity of $\mathcal{L}$ (see Section 3.2). Assuming $\mathcal{L}$ is submodular, then acquisition values exhibit diminishing returns w.r.t. the degree of parallelism $q$. As a result, the marginal value for querying a single point $\mathbf{x} \in \mathcal{X}$ upper bounds its potential contribution to any query set $\mathbf{X}$ s.t. $\mathbf{x} \in \mathbf{X}$. Moreover, marginal acquisition functions $\mathcal{L}(\mathbf{x})$ are substantially cheaper to compute that parallel ones (see Figure 1d). Accordingly, we can initialize query sets $\mathbf{X}$ by sampling from $\mathcal{L}(\mathbf{x})$. By doing so we gracefully avoid initializing points in excluded regions, mitigating the impact of acquisition plateaus.

In our experiments we observed consistent performance gains when using this strategy in conjunction with most query optimizers. To accommodate runtime constraints, the initialization process was run for the first tenth of the allocated time.

Lastly, when greedily maximizing $\mathcal{L}$ (equiv. in parallel asynchronous cases), "pending" queries were handled by fantasizing observations at their predictive mean. Conditioning on the expected value reduces uncertainty in the vicinity of the corresponding design points and, in turn, promotes diversity within individual query sets [15]. To the extent that this additional step helped in our experiments, the change in performance was rather modest.

## B.3 Extended Results

Additional results for both $q$-UCB (Section 3.1) and incremental form $q$-EI (Section 3.2) are shown here. These experiments were run under identical conditions to those in Section 4.

**Parallel UCB** We set confidence parameter $\beta = 2$. Except for on the Levy benchmark, $q$-UCB outperformed $q$-EI, and this result also held for both Branin-Hoo and Hartmann-3 (not shown).

**Incremental $q$-EI** We tested performance using $m \in \{16, 32, 64, 128\}$ states. At the end of the first round of greedy selection $k = 1$, $m$ outcomes $y_1^{(i)} \sim p(y_1 | \mathbf{x}_1, \mathcal{D})$ were fantasized, producing $m$ distinct fantasy states $\mathcal{D}_1^{(i)} = \mathcal{D} \cup \{(\mathbf{x}_1, y_1^{(i)})\}$. At all other steps $k \in [2, q]$, a single outcome was fantasize for each state such that the number of states remained constant. Additionally, fantasized outcomes were never resampled.

Figure 8 compares results obtained when greedily maximizing incremental $q$-EI (with $m = 16$ to those obtained when greedily maximizing joint $q$-EI (as discussed in Section 4). In contrast with the larger body of results, CMA-ES combined with incremental $q$-EI outperformed gradient-based optimization for higher dimensional acquisition surfaces.

**Figure 6:** Average performance of different acquisition maximizers on synthetic tasks from a known prior, given varied runtimes when maximizing Monte Carlo $q$-UCB. Reported values indicate the log of the immediate regret $\log_{10}|f_{\max} - f(\mathbf{x}^*)|$, where $\mathbf{x}^*$ denotes the observed maximizer $\mathbf{x}^* \in \arg\max_{\mathbf{x} \in \mathcal{D}} \hat{y}$.

**Figure 7:** Average performance of different acquisition maximizers on black-box tasks from an unknown prior, given varied runtimes when maximizing Monte Carlo $q$-UCB. Reported values indicate the log of the immediate regret $\log_{10}|f_{\max} - f(\mathbf{x}^*)|$, where $\mathbf{x}^*$ denotes the observed maximizer $\mathbf{x}^* \in \arg\max_{\mathbf{x} \in \mathcal{D}} \hat{y}$.

**Figure 8:** Average performance when greedily maximizing joint vs. incremental forms of $q$-EI.

## Footnotes

[4]This definition comes directly from the standard integral identity [25]: $\int_0^b xe^{-q^2x^2}dx = \frac{1-e^{-q^2b^2}}{2q^2}$.

[5]Due to the stochastic nature of the mean updates, the number of posteriors grows exponentially in $q$.

[6] $\ell_{\mathrm{EI}}$ is often written as the improvement of $\max(\mathbf{y})$; however, these two forms are equivalent.