[Reviews · NeurIPS 2018]

Reviewer 1



The authors propose unifying techniques used in optimizing the qEI and qKG acquisition functions and extend them to most, if not all, acquisition functions of interest in Bayesian optimization, extending these acquisition functions to the batch Bayesian optimization setting in so doing. Since all but one acquisition function rely on the max, which is shown to be submodular, the authors also argue that batch Bayesian optimization can be achieve by greedy construction of the batch leads to an almost exact batch (where "exact" refers to the batch that would optimize the batch acquisition function). In certain parts this paper is slightly sloppy, in particular: - on line 117, where new variables are used without introduction; - in equation (4), which should explicitly include the derivatives of \mu and \Sigma, or at least package them as \theta to avoid needless confusion; - Table 1 features many acronyms which are not introduced until much later, and the last two columns are obscure and not explained in the caption; - or the entire subsection starting at 185, which is presently quite unclear. Though the experimental setting seems fair and the results promising, it would be nice to compare to the performance when using the state-of-the-art method of optimizing q-EI, which I suspect---but could be wrong---is not using random search or CMA-ES, given the differentiability of 1-EI.

Reviewer 2



Summary: This paper proposes a new way of solving the problem of optimizing the acquisition function in Bayesian optimization. The idea is to look at most common acquisitions as the optimization of an expectation of a function, which can be reformulated using the 'reparametrization trick' and optimized using Monte Carlo. The papers shows the reformulation of various acquisitions and presents some interesting experimental results. Strengths and weaknesses of the submission: The paper is really well written and it attacks a problem that affects seriously the performance of new Bayesian optimization methods. The approach is very original and well illustrated and I think that this approach has the potential to become a standard practice in Bayesian optimization alternatively to current approaches that are not even well formalized (each BayesOpt library does it it in its own way). In my opinion this paper brings some foundational aspects to a part of the the optimization loop that has been ignored by the community, in general. Quality: this is a high quality work both because the idea is new but also because the paper is very nicely written. Clarity: the paper is very clear Originality: The method to reparametrize the acquisition is know but its application to this domain is new. Significance: the method is significant for the BayesOpt community in particular and for the ML community in general (given that these techniques are becoming part of the standard toolkit of the ML practitioner).

Reviewer 3



Within the framework of Bayesian optimization, this paper studies the differentiability of Monte Carlo acquisition functions and the submodularity of some acquisition functions. The authors consider to differentiate sample-path to optimize acquisition functions and using submodularity to guarantee the near optimality of greedy approaches. The paper is well written and easy to follow. The efforts on reparameterizing several widely used acquisition functions for differentiability and submodularity in this paper should be recognized. The contributions on methods: differentiating acquisition functions and paralleling submodular acquisition functions are solid, and incremental. Similar ideas have been studied in the past few years. Experiments are well designed and clearly presented. It is a solid paper with some lack of novelty and significance.